# Electromyographic Characteristics of Postactivation Effect in Dopamine-Dependent Spectrum Models Observed in Parkinson’s Disease and Schizophrenia

**DOI:** 10.3390/biomedicines12061338

**Published:** 2024-06-17

**Authors:** Alexander Meigal, Liudmila Gerasimova-Meigal, Anna Kuzmina, Elena Antonen, Alexandra Peskova, Mark Burkin

**Affiliations:** 1Department of Physiology and Pathophysiology, Petrozavodsk State University, 33, Lenina Pr., 185910 Petrozavodsk, Russia; gerasimova@petrsu.ru (L.G.-M.);; 2Department of Neurology, Psychiatry and Microbiology, Petrozavodsk State University, 33, Lenina Pr., 185910 Petrozavodsk, Russia; 3Republican Psychiatric Hospital, 4, Bol’nichnyy Gorodok, 186131 Matrosy, Russia

**Keywords:** dopamine, postactivation effect, muscle tone, electromyography, Parkinson’s disease, schizophrenia

## Abstract

This study aimed to test the hypothesis that the postactivation effect (PAE, involuntary normal muscle tone) is modified by dopaminergic mechanisms. The PAE was tested with surface electromyography (sEMG) in the “off medication” phase in participants with Parkinson’s disease (PD_off_) and in the “on medication” state in participants with schizophrenia (SZ_on_), which modeled hypodopaminegic conditions, and in participants with PD “on medication” (PD_on_) and in participants with SZ “off medication” (SZ_off_) state which modeled the hyperdopaminergic conditions. Healthy age-matched participants constituted the control group (HC, *n* = 11). In hyperdopaminergic models, PAE was triggered in 71.3% of participants in SZ_off_ and in 35.7% in PD_on_ conditions. In the hypodopaminergic models, PAE was triggered in 12% in SZ_on_ and in 21.4% in PD_off_ conditions. In the HC group, PAE was present in 91% of participants. In the HC and PD groups, the mean frequency and correlation dimension of sEMG at PAE was higher than that during voluntary isometric contraction. In conclusion, in hypodopaminergic models, PAE triggering was inhibited. The manifestations and EMG characteristics of PAE in people with PD or SZ may indicate dopaminergic dysfunction.

## 1. Introduction

In the normal nervous system, the neurotransmitter dopamine is involved in the regulation of muscle tone. For example, in rat models, the injection of dopamine into the subthalamic nucleus increases muscle tone threefold [1], which is mediated by dopamine D1 and D2 receptors [2]. The direct application of dopamine on somatic motor neurons in rats triggers muscle tone through D1 receptors [3]. In addition, dopamine plays a central role in the pathogenesis of Parkinson’s disease (PD) [4] and schizophrenia (SZ) [5], pathologies that are characterized by disordered muscle tone. Specifically, in PD, dopamine depletion in the basal ganglia leads to increased resistance to passive joint movement, referred to as muscle rigidity [6]. In SZ, dopamine synthesis in the brain is, in contrast to that in PD, abnormally elevated [7,8], which is often associated with catalepsy, a state of increased muscle tone [9]. Finally, dopamine therapy in PD in 20% of patients results in psychosis [10], while one-third of patients exposed to antipsychotic antidopamine (neuroleptic) therapy develop drug-induced parkinsonism [11]. Altogether, PD and SZ may represent opposite extremes of a dopamine-dependent spectrum, i.e., hypo- and hyperdopaminergic states, as in apathy and impulsivity [12].

The neuromuscular organization of the weak voluntary isometric flexion of the elbow joint, studied using the nonlinear parameters of surface electromyography (sEMG) has features specific to PD [13] and SZ [14], distinguishing them form healthy controls, and is modified by antiparkinsonian therapy [15] and antipsychotic therapy [14], respectively. However, this specificity of sEMG to PD is attenuated with increasing load on the forearm [13]. It would be interesting to know whether these specific sEMG features are present in involuntary muscle tone in participants with PD and SZ. This can be studied using a postactvation effect (PAE) model.

PAE, also referred as the “Kohnstammeffekt” (“Kohnstamm phenomenon”) or aftercontraction phenomenon, represents an involuntary muscle tone (or tonic automatism) that occurs in skeletal muscle following its voluntary isometric contraction [16,17,18,19,20]. In a sense, PAE resembles catalepsy, a state of involuntary postural activity associated with immobility [9] or tonic immobility in animals, birds, and humans [21]. Interestingly, Oskar Kohnstamm described PAE with the term “Katatonusversuch” (a “test for catatonia” in German) [16]. Schizophrenia is often associated with catalepsy [20], which makes it interesting to study PAE in participants with SZ.

PAE is triggered by the poorly identified “tonogenic” centers of the brain, presumably the reticular formation, the cerebellum, and the parietal cortex [17,18,19,20]. Although PAE is triggered in response to the proprioceptor activity of contracting muscle, it cannot be considered a reflex [19]. In healthy people, PAE is characterized by (i) bilateral synchronicity, since it begins and ends simultaneously in the right and left homonymous muscles; (ii) the firing rate of motor units during PAE being 2–3 impulses/s lower than during voluntary isometric contraction of the same amplitude of EMG [22]; (iii) electrical muscle stimulation being unable to induce PAE [19]; and (iv) PAE “switching” to a previously inactive muscle in response to postural adjustments, e.g., head tilt [19]. PAE is presumably generated without the preparation of a “motor command” or “sensory copy”, since it originates from the reticular formation and in this sense resembles a “passive” movement, that is, a movement performed with the help of an external force, which does not need a preparatory stage [18].

Thus, PAE represents a kind of “pure” involuntary but normal muscle tone, not “contaminated” by neural inputs about motivating, planning, and programming of movements, as with active intentional movement. Due to this, PAE can be considered a suitable muscle tone model with which the influence of various factors on muscle tone can be tested. Indeed, PAE is modulated by temperature [23], ground-based microgravity models [24], and sitting and standing position [19].

Based on the above facts, we asked if (1) a type of normal involuntary muscle tone such as PAE be triggered in hypo- and hyperdopaminergic states and (2) if the sEMG characteristics of PAE provide specific information about PD and SD. We hypothesized that, (1) in hypodopaminergic states, PAE is inhibited, and in hyperdopaminergic states, on the contrary, it is enhanced; and, (2) although PAE differs from voluntary isometric con-traction of the central nervous organization, it may reflect either PD- or SZ-specific features.

To test these hypotheses, we examined PAE features (duration and sEMG characteristics) in participants with PD in the off-medication (PD_off_) and on-medication (PD_on_) phases, as well as in participants with SZ with (SZ_on_) and without antipsychotic treatment (SZ_off_). The PD_off_ and SZ_on_ states were used to model the hypodopaminergic state (D−), and the PDon and SZ_off_ states were used to model the hyperdopaminergic state (D+). A control group of age-matched healthy participants presumably represented the optimal dopaminergic state (D). In addition, we aimed to study the correlation of PAE characteristics with the clinical status of people with PD and SZ.

## 2. Materials and Methods

### 2.1. Participants

A total of 11 healthy people in the control group, 14 participants with PD, and 39 participants with SZ of both sexes were included in the study with their informed consent, and this study’s protocol was approved by the Ethics Committee of the Ministry of Health of the Republic of Karelia and Petrozavodsk State University (Protocol No. 38, 10 March 2023). Anthropological data on healthy controls (HCs) and participants with PD and SZ are presented in Table 1. For all participants, noninclusion criteria such as no history of stroke, myopathies, brain traumatic injury, or orthopedic problems were met, as these could have influenced the outcome. The participants with SZ were in general younger than those with PD. Therefore, the average age of the participants in the HC group was intermediate.

All participants with PD were clinically examined by a highly qualified neurologist and attended the trial twice: during the off- and on-medication phases. The clinical status and treatment of participants with PD are presented in Table 2. Immediately after the trial, participants in the PD_off_ phase took their anti-PD therapy. In the on-medication phase, they were at the peak of their anti-PD drug’s effect, which is two hours after taking it, on a different day. Anti-PD treatment included levodopa/carbidopa, pramipexol, piribedil, amantadine, and rasagiline in various dosages and combinations. Levodopa equivalent daily dose (LEDD) [25] was calculated (see Table 2). According to Yahr and Hoehn, all patients with PD, except one, had stage II disease. In the on-medication phase, participants with PD had higher scores on the Unified Parkinson’s Disease Rating Scale (UPDRS) and higher scores for tremor, rigidity, and akinesia (see Table 2).

All participants with SZ were clinically examined by a highly qualified psychiatrist. They were categorized into two groups: (1) those who did not take antipsychotic therapy (*n* = 14, SZ_off_) and were thus modeled in the hyperdopaminergic state and (2) those who took it (*n* = 25, SZ_on_) and were thus modeled in the hypodopaminergic state (Table 3). Of the 14 participants in group SZ_off_, 13 (92.8%) were in remission. In the SZ_on_ group, participants in the remission and psychotic stages were almost equal in number and were taking the typical neuroleptic antipsychotics chlorpromazine, zuclopenthixol, and haloperidol (4 participants); or atypical antipsychotics risperidone, olanzapine, quetiapine, clozapine (8 participants), and a combination of two or three of these drugs (13 participants). The SZ_on_ group had significantly higher scores on the Positive and Negative Syndrome Scale (PANSS) [26], which is used for psychometrics in participants with SZ. For participants with SZ, an additional exclusion criterion was met, namely, the use of non-neuroleptic psychotropic drugs (tranquilizers, antidepressants).

### 2.2. Induction of Postactivation Effect

PAE was triggered in the deltoid muscles by a voluntary isometric contraction (VIC)—abduction of the arms to fixed, vertical, and hard (iron) posts (distance between posts = 95 cm) with the back of the hands for 60 s. The angle at the shoulder joints was approximately 40 ± 5° depending on the height of the participant. The contraction force was approximately half the maximum force to avoid rapid fatigue. This voluntary effort and its duration are considered the threshold for achieving sustained deltoid PAE [27]. The required level of VIC was determined in advance by the sEMG amplitude during the maximum voluntary contraction. In most trials, participants were able to maintain fairly stable sEMG throughout the entire period of voluntary activity. If the sEMG amplitude tended to increase or decrease during voluntary activity, the participant was asked to adjust the effort. At the end of the VIC, the participants, on command, relaxed the deltoid muscles and took a small step forward so that the stands did not interfere with the abduction of the arms. After relaxation, the onset of PAE occurred within 1–7 s in the form of a distinct abduction of the arms to the sides, which coincided with the appearance of electrical activity on the sEMG. PAE was studied in a lighted room with the eyes open, since opening and closing the eyes can cause PAE to switch on and off in antagonist muscles [19].

### 2.3. Electromyography

For sEMG, disposable self-adhesive pregelled silver chloride electrodes (M-00-S type, Ambu^®^, Blue Sensors-M, Ballerup, Denmark) were used. The skin at the electrode position was prepared with 70% ethanol to improve its contact with the skin and conductivity of the sEMG signal. The electrodes were placed side by side (center-to-center distance of 32 mm), along the muscle fibers, over the middle portion of the deltoid muscle, bilaterally. The sEMG signal was obtained using a Neuro-MVP-8 device (Neurosoft LLC, Ivanovo, Russia). The frequency band was 5–1000 Hz; the ADC sampling frequency was 20,000 Hz. During the trial, the power “notch” power filter was disabled, electrical devices in the study room were turned off, and the EMG device was powered from its own battery to avoid powerline interference. All sEMG records were visually inspected to exclude records with nonstationary signals and powerline interference, defined by a characteristic peak at 50 Hz and multiple frequencies (100, 150 Hz, etc.) in the sEMG frequency spectrum. Several sEMG records from participants with SZ were discarded due to powerline 50 Hz inference. Though electromyography was conducted with a battery-supplied EMG device, the locations of the trial were not totally secured from powerline influence.

### 2.4. sEMG Parameters

To characterize the sEMG, the mean frequency of the spectrum (MNF, Hz) was calculated with the software of the electromyograph. The duration of PAE was determined manually using a marker on the electromyograph screen from the onset to termination of the electrical activity in the sEMG recordings. In addition, the fractal and correlation dimensions were extracted from the sEMG signal. The value of the fractal dimension varies from 1 to 2 and indicates the “density” of the signal, that is, the ability of self-similar events in the recording (peaks and turns) to “fill” the surface [28]. The value of the correlation dimension characterizes the complexity of a signal and describes the number of regulators (differential equations) generating the signal [29]. To extract the values of the fractal (D) and correlation (D_c_) dimensions, the sEMG signal was first saved as a text file (*.txt) in a form of time series and then uploaded into FRACTAN 4.4 © software (Institute of Mathematical Problems of Biology RAS, Pushckino, Russia). Then, segments with a length of 80,000 samples (4 s) were selected for analysis in the FRACTAN 4.4 © program. This program computes the Hurst exponent (H) and D, where
D = 2 − H

Samples with a stationary sEMG signal were obtained within the last 10 s of voluntary activity and 10 s after the onset of PAE (Figure 1), so that the effect of fatigue should have been approximately the same during VIC and PAE.

### 2.5. Statistics

sEMG parameters (MNF, D, D_c_) were analyzed with SPSS 21.0 Statistics (SPSS, IBM Company, Natick, MA, USA). Comparisons between VIC and PAE were made with the help of a nonparametric signed-rank Wilcoxon test to reveal differences in sEMG parameters between VIC and PAE in each of the studied groups (PD_on_, PD_off_, SZ_on_, SZ_off_, HC). The nonparametric test was used due to the low incidence of PAE-positive participants in groups PD_on_, PD_off_, and SZ_off_. The differences among the studied groups (PD_on_, PD_off_, SZ_on_, SZ_off_, HC) were assessed with ANOVA for each sEMG parameter (MNF, D, D_c_), separately for VIC and PAE. The duration of PAE was compared among the studied groups and with ANOVA. The correlations between UPDRS-III (Motor Part) and major symptoms (tremor, rigidity, akinesia) scores with duration of PAE were determined with Spearman’s coefficient.

## 3. Results

### 3.1. sEMG Patterns of PAE in the Studied Groups

In healthy controls, PAE was present in 10 out of 11 (90.9%) participants in the form of involuntary abduction of the arms following the VIC. PAE was synchronous on both sides, which means that the onset and offset of PAE took place at the same time in the right and left deltoid muscles. In only one participant, PAE lasted 3 s longer on the left side. A typical sEMG pattern of PAE in a healthy participant is presented in Figure 1.

In participants with PD, PAE was identified visually and electromyographically in 5 of 14 (35.7%) participants in the on-medication phase. Three participants (21.4%, all of those five participants who had PAE in the on-medication phase) had PAE in the off-medication phase. Unlike the controls, the duration of PAE in participants with PD differed in the right and left deltoid muscles in three of five participants with PD (Table 4, Figure 2). 

The duration of PAE of the right deltoid muscle correlated with the UPDRS-III scores (r = 0.85, *p* = 0.014), rigidity (r = 0.88, *p* = 0.047), tremor (r = 0.91, *p* = 0.31), and akinesia (r = 0.91, *p* = 0.030) but not with the UPDRS-I or UPDRS-II scores. Thus, the higher the UPDRS-III scores, the longer the duration of the PAE in the right deltoid muscle. No correlation of duration with LEDD was found (r = 0.56, *p* = 0.32). The low incidence of PAE in the PD_off_ group precluded statistical comparison between the on- and off-medication phases and left- and right-side symptoms.

In the SZoff state, PAE was triggered in 10 of 14 participants, while in the SZon state, only in 3 of 25 participants (see Table 4, Figure 3). In three SZ participants, the duration of PAE slightly differed between the sides of the body. The low incidence of PAE in the SZon group precluded statistical comparison between the on- and off-medication states.

### 3.2. sEMG Parameters of PAE in the Studied Groups

The results are presented in Table 5 and Table 6. Since the values of the sEMG parameters did not differ between the right and left deltoid muscles, the values of the sEMG parameters from both sides were pooled.

In the HC group and in both the PD_on_ and PD_off_ states, the MNF during PAE was higher (*p* < 0.05) than during VIC. During VIC, the MNF was significantly higher in the HC group than in the PD_on_ group (Table 5). In the SZ_off_ group, the values of the MNF did not differ between VIC and PAE. 

The nonlinear sEMG characteristics of PAE are presented in Table 6. The fractal dimension of sEMG varied within a narrow range from 1.72 to 1.79, and it did not differ among the studied groups (see Table 6). The correlation dimension varied around the value of 4.0 in healthy controls in both VIC and PAE (see Table 6). In PD states, during VIC, the correlation dimension decreased to values of 3.5–3.8; during PAE, it slightly, though significantly, increased. In the SZ groups, the correlation dimension did not differ from HC, neither in VIC nor in PAE.

## 4. Discussion

Muscle tone is recognized as a dopamine-dependent phenomenon. In the hyperdopaminergic, state muscle tone is enhanced, and, in the hypodopaminergic state, correspondingly, it is decreased. Dysregulation of the dopaminergic system in the central nervous system leads to abnormalities in muscle tone, such as rigidity in people with PD and catatonia in people with SZ. The PAE phenomenon is an involuntary muscle tone, which is primarily generated in brainstem structures. The original hypothesis in this study was that (1) PAE may be more readily triggered in hyperdopaminergic conditions than in the hypodopaminergic ones, and (2) PAE may represent signatures of PD and SZ on sEMG.

### 4.1. The Incidence of PAE in PD and SZ

This study showed that normal tonic automatism such as PAE is triggered in participants with motor (Parkinson’s disease) and mental disorders (the schizophrenia spectrum disorders), in the pathogenesis of which dopaminergic mechanisms play an important role. The incidence of PAE in the hyperdopaminergic models was indeed higher than that in the hypodopaminergic ones. 

The incidence of PAE in the PD_off_ and PD_on_ states was overall lower (20–35% PAE-positive participants with PD) than in middle age and older controls (91.1% of PAE-positive participants) and young healthy participants (90%) [23,24,30]. This allows us to presume that the low incidence of PAE in the PD_on_ state may rather indicate a hypodopaminergic condition than an optimal dopaminergic or hyperdopaminergic condition. In support of this, in the PD_on_ state, the UPDRS-III score was still high (average 21 vs. 31 in the PD_off_ state, which is a 33% difference). Therefore, the original categorization of the PD_on_ state as a hyperdopaminergic state in this study is likely controversial. Therefore, the incidence of PAE probably just reflected the clinimetric condition in the participants with PD regardless of the medication phase.

A much bigger difference between the on- and off-medication states was documented for SZ. A total of 71.3% patients were PAE-positive in the SZ_off_ group, which is close to the proportion in the group of healthy participants, and only 12% in the SZ_on_ group. This indicates that antipsychotic antidopaminergic therapy with neuroleptics may indeed modulate PAE in SZ. On the other hand, in participants with SZ, the hyperdopaminergic state was best modeled with the off-medication state in the psychotic phase. However, only one such participant was found. Therefore, we modeled that state in the remission phase. In future studies, we plan to recruit a separate group of participants in the off-medication state in the psychotic phase to better model the hyperdopaminergic condition.

Overall, one of the findings of this study is that normal involuntary tonic automatism, such as PAE, is dopamine-dependent in humans and is likely inhibited by the hypodopaminergic condition.

### 4.2. The Pattern of PAE

A difference in the duration of PAE of the left and right deltoid muscle was the characteristic of three of the five PAE-positive participants with PD, unlike the healthy controls, in which the duration PAE on both sides was the same. In healthy participants, PAE can differ between sides when the head is laterally tilted. More specifically, PAE is increased on the contralateral side and decreased on the ipsilateral side, in accordance with the vestibular reflex [19]. In many participants with PD, posture is disordered, forming a specific postural instability and gait disorder (PIGD) phenotype or a subtype of PD [31]. However, the PAE-positive participants in this study mostly had a tremulous phenotype of PD and their head was not visually bent to the right or left. Symptom laterality is specific to PD [32]. Thus, the origins of the asymmetry in the duration of PAE in subjects with PD require further exploration, because the low incidence of PAE in PD precluded an in-depth analysis of the association between symptom laterality and PAE asymmetry. The duration of PAE in participants with PD proportionally depended on the severity of the motor disorders in PD (the UPDRS-III score). The correlation with the scores of UPDRS on each side of the body with PAE duration needs further exploration. 

In some participants with SZ, PAE duration also differed between the left and right side. Altogether, the synchronicity of PAE onset and offset in homonymous muscles of the left and right side of the body can be regarded as signature of the normal state of the motor system. The low incidence of PAE-positive participants in the PD group did not allow conducting an analysis of PAE duration per side.

In addition, no one participant with PD presented an “oscillatory” pattern of PAE, which is the characteristic for approximately one-third of healthy young participants [23,24,25]. This finding indirectly agrees with those of Selionov et al. [33,34] and Solopova et al. [35], which showed that involuntary oscillatory activity (air-stepping) can be triggered by tonic stimuli, including the Kohnstamm phenomenon, in less than 10% of participants with PD in contrast to 60–75% of healthy participants.

PAE exerts notable emotional (“surprising”) effects on participants when they experience the feeling of the “levitation” of their arms (or a feeling of “passive” or “external” motion of arms) [17,18,19]. Due to this, some participants tried to interfere with PAE by involuntary inhibiting or, in contrast, prolonging it. Thus, PAE probably has an ideomotor aspect (unconscious interference with PAE), which must be considered when studying it, especially in subjects with neurologic and psychiatric pathologies. 

### 4.3. sEMG Characteristics

In healthy controls, MNF and the correlation dimension of sEMG during PAE were slightly (by 2–3 Hz) though significantly higher than during VIC. In the study by Meigal et al. [24], an increase in the MNF during PAE was evidenced for younger healthy participants. However, this increase in younger participants was much more notable (by 10 to 15 Hz) [24]. The correlation dimension in healthy controls during PAE was higher by 0.3 (to 4.2) in comparison with during VIC (around 3.9). In comparison, in younger participants, the correlation dimension was equal (close to 4.0) during VIC and PAE [31]. In older adults during voluntary isometric effort, the greater degree of motor unit synchronization and fluctuations in force production [36] are characteristic. This prompts that neuromuscular organization of PAE is rather invariant throughout normal ageing, but this needs further verification.

In subjects with PD, the MNF and correlation dimension were generally less than those of the healthy controls during both VIC and PAE, which can be explained by the “admixture” of recurrent fragments in sEMG, presumably associated with tremor-related oscillations in sEMG [37]. Still, the increase in the correlation dimension in participants with PD during PAE indicates that such an “admixture” becomes less expressed during PAE. The phenomenon of a shift in the sEMG characteristics in the direction of normal values in participants with PD during gradually growing voluntary isometric contraction was already demonstrated [13]. Thus, the sEMG signal in PD participants during PAE is more “normal” than during voluntary contraction. This suggests that the “tonogenic” neural structures involved in the generation of PAE, primarily reticular formation, are less affected by the PD pathology than the basal ganglia.

The participants with SZ presented signatures of the disease neither during voluntary activity nor during PAE. That finding agrees with the fact that participants with schizophrenia in remission are practically undistinguishable from healthy participants using sEMG parameters [13].

The fractal dimension of sEMG was surprisingly uniform among the studied groups and medication conditions, which may reflect an identical timing structure of the sEMG signal and the absence of profound pathology either on the central or neuromuscular level of the motor system.

### 4.4. Limitations and Future Directions

Several limitations of this study and directions for future studies can be identified.

First, in participants with SZ, the hyperdopaminergic state was best modeled with the off-medication state in the psychotic phase. However, only one such participant was recruited to this study. Therefore, the hyperdopaminergic state was modeled with the remission phase, which is less specific for the hyperdopaminergic state. In future studies, we plan to recruit more participants in the off-medication state and in the psychotic phase to better model the hyperdopaminergic condition.

The low incidence of PAE-positive participants in the PD_off_ and SZ_on_ groups, originally categorized as the hypodopaminergic condition, did not allow us to fully characterize them and compare them with the opposite medication state/phase. More measurements should be obtained to complete these groups. In our earlier study, we found that the sEMG characteristics in the participants with SZ depended on the kind of antipsychotic therapy (typical or atypical) [13], which can also be examined in future studies during PAE.

The participants with PD and SZ clearly differed in age: participants with SZ were younger by an average of 10–15 years. Due to this, the group of healthy controls comprised participants within a wide age range. In future studies, two separate groups of healthy controls with average ages of around 45 years and 65 years must be formed.

The offset point of PAE in some participants was not clearly identifiable, which can be regarded as a limitation of this study. In future studies, we recommend adding the goniometric signal to better identify both the beginning and termination of PAE.

In addition, PAE is most readily triggered and visually explicit in the deltoid muscles. However, in participants with PD, muscle tone disorder (muscle rigidity) is the characteristic of flexor musculature, for example, the *m. biceps brachii*. Commonly, participants with PD have a bent posture (camptocormia). Muscle rigidity in PD, according to the UPDRS-III, is tested at the elbow joint not the shoulder. Also, tremor, another cardinal motor symptom, is usually present in the distal joints of the body (hands and forearms). Therefore, in future studies, it would be interesting and logical to study PAE patterns and sEMG characteristics in the *m. biceps brachii*.

Finally, PD is characterized by a specific pattern of motor unit activity during isometric contraction: discharges with intermittent (doublets) and variable interspike intervals [38,39]. The identification of doublets during PAE in participants with PD would have served a kind of experimentum crucis for understanding the involvement of PAE in PD.

## 5. Conclusions

This study revealed that PAE can be triggered in participants with pathologies and in states modeled in either the hyper- or hypodopaminergic state. In participants with SZ in the off-medication condition and in those with PD in the on-medication condition, which were modeled in the hyperdopaminergic state, the incidence of participants that were PAE-positive was higher (36 to 71%) than in the models of the hypodopaminergic state (12 to 21%), which supports the idea of the dopamine-dependent nature of PAE. The sEMG parameters during PAE in participants with PD shifted in the direction of normal values, though they were still less than those in healthy participants.

The low incidence of PAE in the three studied groups did not allow the conducting of a relevant analysis of the correlation between symptom laterality and PAE. However, the lower incidence of PAE in hypodopaminergic states and the laterality of PAE in PD indicates that it has some diagnostic value. As such, PAE as a model of involuntary tonic automatism seems a promising test for the diagnosis of CNS disorders.

## Figures and Tables

**Figure 1 biomedicines-12-01338-f001:**
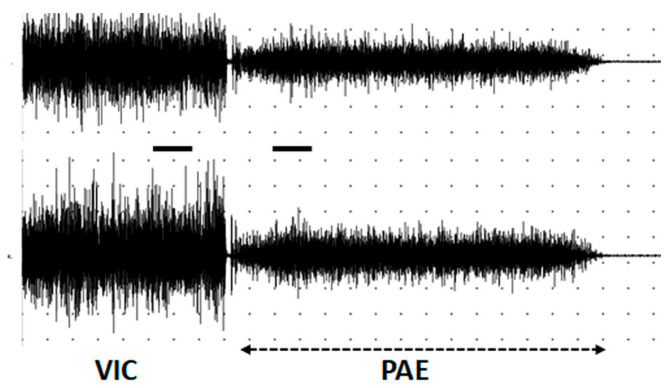
A representative sEMG pattern of postactivation effect in right (**top**) and left (**bottom**) deltoid muscle in a healthy control (58 years). VIC—voluntary isometric contraction; PAE—postactivation effect; calibration—5 s, 500 mkV; horizontal bars—sites of sample extraction.

**Figure 2 biomedicines-12-01338-f002:**
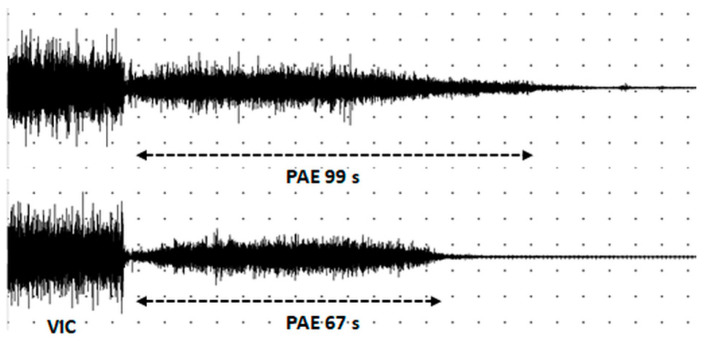
A representative sEMG pattern of postactivation effect in right (**top**) and left (**bottom**) deltoid muscle in a participant with PD in the PD_on_ state (UPDRS-III 27, tremulous form, right-side prevalent). VIC—voluntary isometric contraction; PAE—postactivation effect; calibration—5 s, 500 mkV.

**Figure 3 biomedicines-12-01338-f003:**
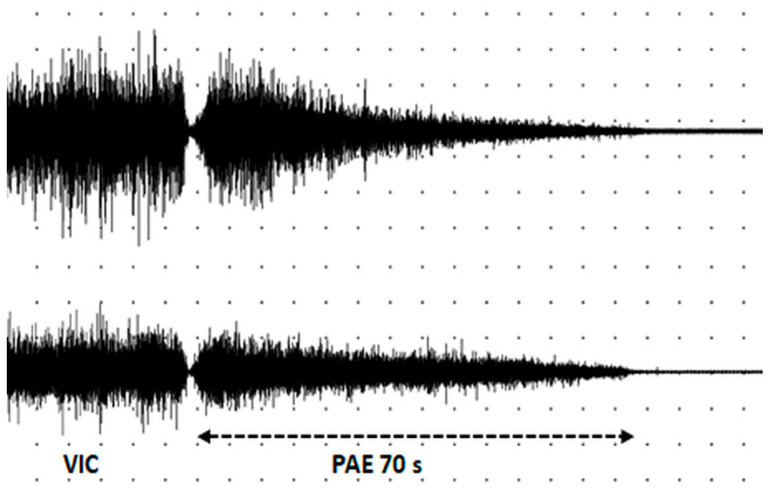
A representative sEMG pattern of postactivation effect in right (**top**) and left (**bottom**) deltoid muscle in a participant with SZ in the SZ_off_ state (remission, 23 years old). VIC—voluntary isometric contraction; PAE—postactivation effect; calibration—5 s, 500 mkV.

**Table 1 biomedicines-12-01338-t001:** Anthropometry characteristic of participants.

Group	N (f, m)	Age (Years)	Height (cm)	Body Mass (kg)	BMI
HC	11 (6, 5)	52 (38; 62)	172 (165; 175)	75 (65; 89)	24.8 (22.5; 28.7)
PD	14 (6, 8)	62 (55; 66)	169 (165; 176)	67 (62; 76)	23 (22; 25.2)
SZ	39 (21, 18)	37 (26; 51)	168 (160; 177)	76 (69; 85)	25.5 (21.9; 31.3)

Results are presented as Me (25%; 75%). Abbreviations: HC, healthy control; PD, participants with Parkinson’s disease; SZ, participants with schizophrenia; BMI, body mass index.

**Table 2 biomedicines-12-01338-t002:** Clinical characteristic of participants with PD in on- and off-medication conditions.

Group	UPDRS-I	UPDRS-II	UPDRS-III	Tremor	Rigidity	Akinesia	LEDD mg/Day
PD_on_	1 (0; 2.7)	6 (4; 9)	21 (15; 25)	1 (0; 3.5)	6 (4; 9)	8 (5; 9)	282 (100; 350)
PD_off_	0.5 (0; 3.5)	11.5 (8; 16)	31 (28; 33) *	3.5 (2.3; 5.5)	7 (6; 8)	11.5 (9; 11.8)	

Results are presented as Me (25%; 75%). * *p* < 0.05, the significance of differences is based on paired Wilcoxon test.

**Table 3 biomedicines-12-01338-t003:** Clinical characteristic of participants with SZ in on- and off-medication conditions.

Group	N (f, m)	Age (Years)	Height (cm)	Body Mass (kg)	BMI	PANSS	Remission	Psychotic Stage
SZ_on_	25 (14, 11)	37 (27; 50)	168 (157; 180)	74.5 (49; 88)	25.5 (21.9; 31.3)	79 (61; 75)	11	14
SZ_off_	14 (7, 7)	44 (25.5; 53)	168 (163; 176)	76 (71; 88)	27 (22.7; 31.5)	68 (69; 99) *	13	1

Results are presented as Me (25%; 75%). * *p* < 0.05, the significance of differences is based on the Mann–Whitney test.

**Table 4 biomedicines-12-01338-t004:** Duration of PAE in PD and SZ participants and healthy controls.

Dopaminergic Condition	D−		D	D+	
Groups	SZon	PDoff	HCs	PDon	SZoff
PAE-positive participants (%)	3/25 (12)	3/14 (21.4)	10/11 (91.1)	5/14 (35.7)	10/14 (71.3)
Duration, right side (s)	(60; 66)	(24; 33)	81 (42; 101)	52 (26; 144)	51 (30; 115)
Duration, left side (s)	(62; 66)	(33; 35)	81 (42; 102)	80 (27; 144)	51 (30; 121)

Results are presented as Me (25%; 75%). Abbreviations: HC, healthy control group; PD, participants with Parkinson’s disease; SZ, participants with schizophrenia.

**Table 5 biomedicines-12-01338-t005:** Mean frequency of sEMG spectrum of PAE in participants with PD and SZ and healthy controls.

Dopaminergic Condition	D−		D	D+	
Groups	SZ_on_	PD_off_	HC	PD_on_	SZ_off_
VIC	56.9 (53.3; 68.3)	54.1 (47.1; 62.6)	67.0 (59.8; 73.6)	50.1 (46.4; 60.3) #	80.3 (72.2; 81.6)
PAE	(60; 62.3)	66.9 (54.9; 70.9) *	69.5 (63.,5; 75.7) **	61.1 (51.1; 64.1) * #	78.7 (74.1; 85.4)

Results are presented as Me (25%; 75%). Abbreviations: HC, healthy control group; PD, participants with Parkinson’s disease; SZ, participants with schizophrenia; VIC, voluntary isometric contraction; PAE, postactivation effect. #, *p* < 0.05, the significance of differences from HCs is based on ANOVA. *, *p* < 0.05; **, *p* < 0.01, the significance of differences between VIC and PAE is based on paired Wilcoxon test.

**Table 6 biomedicines-12-01338-t006:** Fractal dimension of sEMG signal in participants with PD and SZ and healthy controls.

Dopaminergic Condition		D−		D	D+	
Groups		SZon	PDoff	HC	PDon	SZoff
Fractal dimension	VIC	1.82 (1.78; 1.84)	1.79 (1.75; 1.82)	1.79 (1.72; 1.84)	1.78 (1.78; 1.83)	1.80 (1.74; 1.82)
	PAE	(1.72; 1.79)	1.81 (1.75; 1.82)	1.76 (1.73; 1.80)	1.78 (1.74; 1.84)	1.79 (1.74; 1.82)
Correlation dimension	VIC	4.12 (3.61; 4.58)	3.66 (2.61; 3.84)	3.90 (3.80; 4.20)	3.66 (3.25; 3.92)	3.93 (3.64; 4.18)
	PAE	(3.91; 3.99)	4.00 (2.61; 4.33) * #	4.17 (4.05; 4.70) **	3.89 (3.27; 4.21) * #	3.96 (3.73; 4.08)

Results are presented as Me (25%; 75%). Abbreviations: HC, healthy control group; PD, participants with Parkinson’s disease; SZ, participants with schizophrenia; VIC, voluntary isometric contraction; PAE, postactivation effect. #, *p* < 0.05, the significance of differences from HC is based on ANOVA. *, *p* < 0.05; **, *p* < 0.01, the significance of differences between VIC and PAE is based on paired Wilcoxon test.

## Data Availability

All other data that support the findings of this study are available from the corresponding authors on reasonable request.

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
