# Peer review of "Electromyographic Characteristics of Postactivation Effect in Dopamine-Dependent Spectrum Models Observed in Parkinson’s Disease and Schizophrenia"

_biomedicines, 2024, doi:10.3390/biomedicines12061338_

Round 1
Reviewer 1 Report
Comments and Suggestions for Authors
Due to the limitations of this study, the conclusion and interpretation of the findings must be cautious. the authors are asked to elaborate on this point. Please also add an alternative method that could have been used to test the hypothesis. the authors have also indicated that the findings here can be applicable to the clinical application of CNS disorders diagnosis. It is suggested that this remains only limited to the conditions studied here and not general CND disorders. As the authors have also mentioned, the generalizability of findings is low due to the limitations.
Related to the location of the measurements, please add whether the site is an optimal or standard r based on the authors' preliminary data. Do the authors expect that changing the location might result in a dramatic change in the findings?
Reviewer 2 Report
Comments and Suggestions for Authors
Overall, a study appears to be warranted. The design was controlled and the manuscript was well-organized. There is a plethora of word choice and grammar errors that need to be corrected. See specific comments in the pdf.

Several errors need correction.
Reviewer 3 Report
Comments and Suggestions for Authors
The study aimed to test the hypothesis that the postactivation effect (PAE, an involuntary normal muscle tone) is modified by dopaminergic mechanisms. PAE was tested with surface electromyography (sEMG) in the “off medication” phase in subjects with Parkinson’s disease (PDoff) and the “on medication” state in subjects with schizophrenia (SZon), which modeled hypodopaminergic conditions, and in PD subjects “on medication” (PDon) and SZ subjects “off medication” (SZoff), which modeled the hyperdopaminergic conditions. The article meets the correct conditions for publication as it presents a good literature review, the methodology is correct, and the statistical tests are also appropriate. My only suggestion is regarding the title, where I believe it could indicate the pathologies involved, and this cite : https://doi.org/10.3233/JPD-202067
Round 2
Reviewer 2 Report
Comments and Suggestions for Authors
Appropriate edits were highlighted, however, some additional word selections are advised, particularly eliminating the use of the word subjects. See pdf.

word choice errors need to be corrected.
